# Comparison of Anatomical Preformed Titanium Implants and Patient-Specific CAD/CAM Implants in the Primary Reconstruction of Isolated Orbital Fractures—A Retrospective Study

**DOI:** 10.3390/jpm13050846

**Published:** 2023-05-17

**Authors:** Sebastian Pietzka, Markus Wenzel, Karsten Winter, Frank Wilde, Alexander Schramm, Marcel Ebeling, Robin Kasper, Mario Scheurer, Andreas Sakkas

**Affiliations:** 1Department of Cranio-Maxillo-Facial-Surgery, University Hospital Ulm, 89081 Ulm, Germany; frank.wilde@uni-ulm.de (F.W.); alexander.schramm@uni-ulm.de (A.S.); andreas.sakkas@uni-ulm.de (A.S.); 2Department of Cranio-Maxillo-Facial-Surgery, German Armed Forces Hospital, 89081 Ulm, Germany; markus.wenzel@uni-ulm.de (M.W.); mrclebeling@gmail.com (M.E.); robinkasper@bundeswehr.org (R.K.); marioscheurer@bundeswehr.org (M.S.); 3Institute of Anatomy, Medical Faculty, University of Leipzig, 04109 Leipzig, Germany; kwinter@rz.uni-leipzig.de

**Keywords:** orbita reconstruction, PSI, anatomical preformed orbital implant, CAD/CAM, intraoperative 3D-C-Arm

## Abstract

Background/Aim: Reconstruction of the fractured orbit remains a challenge. The aim of this study was to compare anatomical preformed titanium orbital implants with patient-specific CAD/CAM implants for precision and intraoperative applicability. Material and Methods: A total of 75 orbital reconstructions from 2012 to 2022 were retrospectively assessed for their precision of implant position and intra- and postoperative revision rates. For this purpose, the implant position after digital orbital reconstruction was checked for deviations by mirroring the healthy orbit at 5 defined points, and the medical records of the patients were checked for revisions. Results: The evaluation of the 45 anatomical preformed orbital implant cases showed significantly higher deviations and an implant inaccuracy of 66.6% than the 30 CAD/CAM cases with only 10% inaccuracy. In particular, the CAD/CAM implants were significantly more precise in medial and posterior positioning. In addition, the intraoperative revision rates of 26.6% vs. 11% after 3D intraoperative imaging and the postoperative revision rates of 13% vs. 0 for the anatomical preformed implants were significantly higher than for patient-specific implants. Conclusion: We conclude that patient-specific CAD/CAM orbital implants are highly suitable for primary orbital reconstruction. These seem to be preferable to anatomical preformed implants in terms of precision and revision rates.

## 1. Introduction

The functional and anatomically correct reconstruction of the fractured orbit in craniofacial trauma remains a major challenge. The best possible material for reconstruction is still controversially discussed [1]. The first descriptions of the treatment of orbital floor fractures date back to the 18th century. Initially, autologous bone or tissue grafts were primarily used for this purpose. These are still used today in part and depending on the treating specialist’s discipline. In a further development, polymers, ceramics and plastics were also used. In the further course, resorbable PDS membranes were used, especially for small defects. However, all these materials have their limitations. On the one hand, they are either very voluminous in order to achieve dimensional stability or, on the other hand, they are very thin and flexible but cannot assume a defined geometry.

Another solution was, therefore, the use of titanium grids. These could be customized intraoperatively to the dimension and adapted to the s-shaped geometry of the orbit. This was the gold standard for a long time until the further development of CAD/CAM procedures for the production of individual skull models. Only with these individual CAD/CAM skull models was it possible to individualize the orbital grids preoperatively by bending and cutting them to size. With this, an improved precision could be demonstrated [2]. However, here, too, the lead time for model production was a limiting factor.

Metzger et al. were involved in the development of anatomical preformed titanium meshes. These were developed after evaluating the anatomical structures of the orbit from multiple CT data sets. Corresponding to their analyses, anatomical preformed titanium meshes were offered in two different sizes for both sides of the orbit [3,4,5,6]. When using the anatomical preformed meshes in a cadaver study, Metzger et al. could not demonstrate any disadvantage compared to the orbital meshes individually adjusted on individual CAD/CAM skull models [7]. The big advantage, however, was that these anatomical preformed meshes were available out of the box in the OR without any lead time or further planning.

Different methods were developed to improve the positioning of the titanium mesh intraoperatively. In particular, intraoperative navigation was initially a milestone here and should be mentioned. Navigation enabled immediate radiation-free position control in the operating theatre. The disadvantage, however, was that in addition to the high cost of a navigation system, special registration had to be carried out using defined points. Furthermore, the intraoperative fixation of an optically detectable device by screws on the skull was necessary in order to carry out the navigation. This procedure could be somewhat time-consuming [8].

Another milestone in orbital surgery and reconstruction with titanium meshes was the development of mobile 3D C-arms that can be used intraoperatively. This made it possible for the first time to check the intraoperative position without prior elaborate planning of the navigation, even in acute trauma. The benefit of intraoperative 3D C-arms was also demonstrated in the treatment of complex midface fractures by Wilde et al. [9]. Due to the intraoperative imaging, an additional reconstruction of the orbita was no longer necessary in a large number of cases after the reduction in the zygomatic bone, in contrast to the preoperative planning [10].

Through further development of CAD/CAM technology and, in particular, additive laser melting technology, patient-specific titanium orbital implants could be manufactured from 2014 onwards. For this purpose, the healthy orbital bone was often mirrored on the fractured one in a fully digital workflow, and the titanium grid was computer-assisted designed and computer-assisted manufactured over the defect accordingly. The delivery times for PSI were initially several weeks so this technique was mainly suitable for the secondary reconstruction of extremely complex orbital fractures. Due to changes in the planning and production processes, it is now possible to plan, produce and deliver within 5 working days, depending on the selected manufacturer.

The aim of this study was to compare the precision of the two reconstruction methods, performed titan meshes and patient-specific CAD/CAM meshes. Another aim was to compare the intraoperative applicability and the necessity of intraoperative and postoperative revisions after incorrect positioning of the titanium mesh in relation to the two surgical methods.

## 2. Materials and Methods

For this observational retrospective single-center study, the hospital information system was queried for orbital fractures according to the ICD 10 diagnosis S02.3. We reviewed medical records of all patients who underwent operative orbit reconstruction after traumatic fractures in the clinic of oral and plastic maxillofacial surgery of the German Armed Forces Hospital Ulm between September 2012 and March 2022. Records were retrieved from our hospital electronic database. Ethical approval for this study was obtained from the ethics committee of University Ulm (approval number: 508/20). This study was performed in accordance with the Declaration of Helsinki 1964 and its later amendments (World Medical Association, Declaration of Helsinki). We enrolled patients who fulfilled the following inclusion criteria: (1) Surgical treatment of isolated unilateral orbital floor fractures or combined unilateral orbital floor and orbital wall fractures, (2) pre-operative 3D imaging, (3) intraoperative 3D imaging via 3D-C-Arm or postoperative 3D imaging and (4) OP—report. Exclusion criteria were (1) no preoperative 3D Imaging, (2) no postoperative 3D imaging, (3) combined bilateral orbital fractures, (4) combined fractures of the orbital and the midface and (5) incomplete medical charts.

All surgical procedures were performed either by a board-specialized oral and maxillofacial surgeon or by a resident under supervision. A transconjunctival approach was defined as the standard access to the orbit.

Based on the surgical reports, medical records and intraoperative or postoperative imaging, it was possible to determine whether an anatomical preformed orbital plate (SPOP) or a patient-specific CAD/CAM implant (Orbital PSI) was used.

When the surgical treatment was performed using an anatomical preformed orbital plate, the surgeon selected the implant (DePuy Synthes^®^, Matrix MIDFACE orbital plates™, West Chester, PA, USA) corresponding to the side of the fractured orbit and its defect size intraoperatively (Figure 1).

If, on the other hand, a CAD/CAM manufactured patient-specific plate was used, the preoperative CT data set was transferred preoperatively to one of the PSI manufacturers.

Subsequently, the digital orbital reconstruction was performed by mirroring the unaffected orbit onto the fractured one in cooperation with their medical engineers. The design was then computer-assisted and designed according to the defect and computer-assisted manufactured using the additive laser melting process (Figure 2).

The decision whether to commission the PSI from DePuy Synthes© (Westchster, PA, USA) in cooperation with Materialise© (3001 Leuven, Belgium), KLS© (Tutlingen, Germany) or ReOss© (Filderstadt, Germany) was partly purely random and partly dependent on the production time and capacity of the three companies. There was no personal preference or preference among the performing surgeons based on operational handling. All PSIs were manufactured to the same departmental specifications in terms of shape and design. Any influence of the manufacturer on the accuracy of the implant position or the possibility of implant insertion is, therefore, very unlikely.

Intraoperatively, a 3D C-arm examination was performed after implant placement to check the position of the implant. If a 3D C-arm was not used intraoperatively, a 3D image was taken postoperatively using CT.

Until 2016, reconstruction with PSI was carried out exclusively for particularly pronounced defects and for secondary reconstruction. From 2016 onwards, PSI was also used for primary orbital reconstruction due to the significantly shorter production time. From 2019 onwards, PSI was used the first choice whenever possible for all complex orbital reconstructions whenever a titanium mesh was indicated, and SPOP was only the exception.

### 2.1. Data Collection

One single examiner abstracted all available preoperative and postoperative radiological findings inside the hospital PACS system. Data were collected from patients’ hospital charts, and patients were anonymized before data analysis.

Patients enrolled were subsequently divided into two groups according to the surgical method used for orbital reconstruction: (1) orbital reconstruction via anatomical preformed orbital implant (APOI) in 45 patients and (2) orbital reconstruction via patient-specific CAD/CAM orbital Implant (Orbital PSI) in 30 patients.

The demographic data were determined from the hospital information system. In addition, the following treatment-specific data were collected: (1) duration from trauma to surgery, (2) a number of intraoperative position corrections of the implant after 3D-C-arm control and (3) number of revisions in the context of a second operation in case of persistent complaints (double images and motility restrictions) or due to postoperative CT diagnostics. In the case of (3), the time between primary and secondary surgery was also determined in days.

Due to the retrospective study design, a statement about the functional postoperative course was only possible to a limited extent. Standardized follow-up with ophthalmological findings > 14 days postoperatively was only possible in cases with symptoms. This meant that no statements could be made about the long-term course in this study. In particular, a consistent examination for enophthalmos >6 weeks postoperatively was not documented. Due to the relatively long observation period of >10 years, a consistent follow-up examination of the patients was not performed.

### 2.2. Radiological Evaluation

Further investigation of the 3D imaging was performed after importing the DICOM data into the software Brainlab© iPlan CMF™ 3.0 (Brainlab©, Munich, Germany). In the data set of the preoperative 3D imaging, the facial skull was segmented semi-automatically by the software. Subsequently, the intact orbit was mirrored onto the defect orbit in order to generate the structures of an anatomically correct orbit and also in the areas of the bony defect (Figure 3a,b). Using the “smartbrush function” of the software, the parts of the digitally reconstructed orbit that deviated from the existing bone structure were now touched up. Afterward, the digital orbita, which had been ideally reconstructed for each case, was checked for accuracy by a specialist in maxillofacial surgery together with the examiner according to the four-eyes principle. A classification was also made according to the complexity of the fracture: (1) simple orbital floor fracture, (2) severe orbital floor fracture with clear extension to the dorsal, (3) orbital floor fracture with involvement of the medial orbit and (4) severe orbital floor fracture with complex extension to medial and dorsal. In the next step, the intraoperative/postoperative imaging of the corresponding case was imported into the iPlan CMF™ 3.0 software. This was followed by the fusion and superimposition with the preoperative data set, including the idealized reconstructed orbit (Figure 3c). The software’s “autofusion function” was used for this and, if necessary, optimized manually again.

Subsequently, the position of the titanium grid in the postoperative imaging was compared with the idealized orbit. Deviations were examined and measured at the following 5 points: (1) medial implant placement, (2) lateral implant placement, (3) anterior implant placement, (4) posterior implant placement and (5) central implant placement. The software function was used for measurement, and the largest deviation in the axial, sagittal or coronal layer was drawn in the multiplanar reconstruction. The deviation at the corresponding position was determined in mm (Figure 4 and Figure 5).

If the deviation was less than one mm, it was described as “<1 mm”. A perfect overlay at the defined position was marked with a deviation of 0. The examination was carried out independently of the implant type. Furthermore, the measurement was carried out independently by both the examiner and the specialist in maxillofacial surgery. In case of deviations of more than 0.3 mm between the two evaluators, the case was evaluated again jointly according to the four-eye principle, and the largest detectable deviation was registered. In the case of deviations of less than 0.3 mm between the two examiners, the values from both evaluators were averaged.

### 2.3. Statistical Analysis

Data were centralized in electronic format using Microsoft Excel 2019 software (Microsoft Corporation, Redmond, WA, USA) and analyzed descriptively. Statistical analysis was performed using IBM SPSS^®^ Statistics 26.0 (IBM, Armonk, NY, USA). Metric data were expressed as mean and standard deviation (SD), while nominal data were expressed as frequency and percentage. Descriptive statistics were used to describe baseline patient characteristics. All categorical variables were expressed as absolute values (n) and relative prevalences (%).

## 3. Results

### 3.1. Patient Collective

In the observation period from January 2012 to March 2022, 81 patients underwent surgery for an orbital fracture with titanium meshes at the German Armed Forces Hospital Ulm. This retrospective study included 75 patients who met the inclusion criteria. A total of 6 patients (5 APOI and 1 PSI) could not be included due to a lack of sufficient postoperative 3D imaging and case documentation. A total of 45 patients were treated with an APOI and 30 patients with a PSI. There were more males (*n* = 45; 60%) than females (*n* = 30; 40%) (male to female ratio = 1.5:1). Patient age at the time of surgery ranged from 19 to 87 years, with a mean age of 45.5 years.

The main cause of orbital fracture was brute force (34.6%) and tripping (33.3), followed by recreational accidents (22.7%) and traffic accidents (9.3%). The bicycle was the preferred mode of transport in 85.7% of the group of traffic accidents with orbital fractures. Of the recreational accidents with isolated orbital fractures, 41.2 percent were skiing/winter sports accidents. This can certainly only be explained by the local conditions and the proximity to winter sports areas.

Of the 45 orbital fractures treated with APOI, 24.4% (*n* = 11) had a fracture of the medial orbital wall in addition to an orbital floor fracture. In contrast, in the group of treated orbital fractures with PSI, 36.6% (*n* = 11) had an additional medial orbital wall fracture. There were no additional fractures of the lateral orbita or orbital roof in this patient collective.

The median time between trauma and surgery was 5.33 days in the APOI group (median 4.5, min 0, max 15) and 12.77 days in the PSI group (median 12.5, min 6, max 22). In particular, in the years 2020 to 2022, a consistent time span of less than 14 days between trauma and time of surgery could be shown.

### 3.2. Evaluation of the Intraoperative Revisions and the Secondary Interventions

The evaluation of the intraoperative corrections in the APOI group showed that repositioning was necessary in 26.6% of the cases (*n* = 12) after intraoperative 3D C-arm control. In these 12 cases, the implant had to be repositioned twice in two patients and three times in one patient after an intraoperative 3D C-arm check. In this group, despite intraoperative 3D-C-arm control, 11.1% (*n* = 5) of patients required a second operation in the course of time for persistent complaints due to malposition of the APOI. Secondary revision surgery was performed in all 5 patients within a mean of 4 days (median 5, min 1, max 6 days) due to significant postoperative restrictions of motility and double vision in the central field of the vision. In all these cases, there was a clear elevation of the orbital meshes in the posterior part.

In contrast, the evaluation of the intraoperative reductions after intraoperative 3D C-arm control showed only 13% (*n* = 4) corrections in the PSI group. No correction had to be repeated intraoperatively. Furthermore, no secondary operations were indicated due to PSI mispositioning.

### 3.3. Evaluation of the Postoperative Deviations

The deviation examination was performed in all patients by measuring at the five defined locations between digitally mirrored and reconstructed orbits and the last intra- or postoperative imaging. The deviation was not calculated for the revised implant positions, neither in the position before intraoperative correction nor in the position before two-stage revision surgery.

#### 3.3.1. Deviations APOI Group

The measurement of the postoperative implant position in the APOI group showed a deviation of >1 mm in at least one of the five measured positions in 66.7% of the patients (*n* = 30) (Table 1). In this group, 20% of the patients (*n* = 9) showed a deviation in the medial orbital region. The mean deviation was 3.17 mm (median 2.6). In three patients, there was an overcorrection (max. 6 mm) and in six patients an undercorrection (max. 6 mm). In three of these six patients, the medial orbital fracture with undercorrections of 6 mm, 6 mm and 5.5 mm was not covered by the implant and thus not sufficiently corrected. In the anterior localization, only two patients showed a deviation of 1.9 mm and 2.3 mm in the sense of an undercorrection. No deviations could be determined in the lateral localization. Centrally, there were nine deviations in the APOI group with a mean of 2.36 mm (median 1.9 mm). There were central undercorrections in three cases (max. 6 mm) and overcorrections in six cases (max. 2.8 mm). Most of the deviations in the APOI group were at the dorsal orbital bone. Here, 60% (*n* = 27) of the operated patients had deviations of 2.49 mm on average (median 2 mm). There were 25 overcorrections (max. 6.7 mm) and only 2 undercorrections (max. 6 mm). The mean complexity of the fractures in the APOI group, as determined by the observer, was 2.33 (median 3) with a definition of one for a simple orbital floor fracture to five for an extremely complex multi-wall orbital fracture.

#### 3.3.2. Deviations PSI Group

The medial localization showed only two deviations. One with an overcorrection of 1.1 mm and one with an undercorrection of 2.2 mm (Table 1). Anteriorly, there was only evidence of one overcorrection of 1.1 mm. In the posterior orbital region, there were three deviations with 2.2, 2 and 1.1 mm overcorrection. No deviations were found in the lateral and central positions. All fractures of the medial orbit in this group were covered by the implant and consequently reconstructed. The mean complexity of the fractures in the PSI group, as determined by the observer, was 2.73 (median 3), with a definition of 1 for a simple orbital floor fracture to 5 for an extremely complex multi-wall orbital fracture.

### 3.4. Comparison APOI and PSI Group

In the APOI group, significantly more intraoperative revisions had to be performed than in the PSI group (26.6% vs. 13%) (Table 2). Only in the APOI group were secondary corrections necessary due to the incorrect positioning of the titanium grid and the resulting double vision and motility restrictions (11.1%, *n* = 5).

Furthermore, the APOI group showed a more frequent under-reconstruction of the medial orbit. In the PSI group, on the other hand, all parts of the medial orbital fracture were covered by the PSI and sufficiently reconstructed. The PSI group contained the supposedly more difficult operations. Both the classification by complexity assigned by the investigators was higher (2.73 vs. 2.33), as was the proportion of fractures with additional involvement of the medial orbit (36.6% vs. 24.4%).

## 4. Discussion

We conducted this study in an oral and maxillofacial surgery clinic to look at the influence of two different implant types on reconstruction accuracy in orbital fractures. Furthermore, we wanted to assess the intraoperative applicability in relation to the need for revision.

The anatomically correct and functional reconstruction of the orbital floor and media wall fractures remains a special challenge in surgical treatment in the facial region.

In addition to the challenge of the operation itself, the first difficulty is the indication. In particular, the correct choice between immediate surgical treatment on the one hand and early or even delayed surgical therapy, on the other hand, requires the correct assessment by the surgeon [11,12,13,14,15]. On the contrary, there is a purely conservative therapy approach without surgical intervention. Indications for immediate surgical treatment (within hours) of an orbital floor or media wall fracture are rather rare. This includes infantile trapdoor fractures, persistent oculocardiac reflexes and wide-open soft tissue injuries with direct access to the orbit. For immediate reconstruction, when a titanium mesh is indicated, out-of-the-box solutions are nowadays the only alternative. In certain concomitant injuries, such as retrobulbar hematoma, bulbar laceration or even unstable patients, immediate surgery may also be contraindicated.

Alternatively, early surgical treatment within 14 days may be indicated. The advantage is significantly reduced periorbital swelling, which can facilitate surgical access to the orbit. Within this time, intraocular scar healing seems to be irrelevant, and the delay of surgical therapy is not of negative effect on the patient.

In these cases, not only prefabricated titanium meshes but also CAD/CAM patient-specific implants can be used if the patient presents to the clinic at an early stage and the internal clinical procedures are clearly regulated [16,17,18,19,20,21]. In delayed, secondary orbital reconstruction, patient-specific CAD/CAM implants appear to be advantageous [22,23,24]. In their systematic review of customized products for orbital wall reconstruction, Hartman et al. were able to show a very heterogeneous data situation. For them, there was a slightly positive trend in surgical interventions with patient-specific implants. However, the fact that some of the orbital implants attached to the model were also listed as patient-specific in the studies made it difficult to evaluate them [24]. In a prospective study of 96 orbital fractures reconstructed using PSI, Rana et al. demonstrated good clinical applicability and high precision in reconstructing orbital volume. A 3D analysis with color mapping showed only minor deviations. However, a concrete description of the deviations at individual localizations could not be shown here [20]. The use of intraoperative navigation seemed to allow a further improvement of the positioning. Probst et al. were able to demonstrate high precision for freehand positioning in most cases for PSI for the reconstruction of the orbital floor and/or the medial wall. In contrast to the use of standard implants, the use of a navigation system seems to be of secondary importance here [25]. Furthermore, the positive influence of intraoperative 3D imaging on orbital implant position has been demonstrated in cadaveric studies and in clinical investigations [9,10,26].

This study also evaluated the applicability of intraoperative positioning. Here, a clear advantage seems to lie in the defined geometry of the PSI. Through extension and the creation of a defined contact zone on the lateral orbital wall, correct positioning seems to be significantly easier. Compared to the APOI group, the need for intraoperative corrections after 3D imaging was reduced from 26.6% to 13%. Deviations between implant position and bone apposition have been described as an indication for intraoperative correction in the case of a supposedly clinically good position. According to the evaluation of the surgical reports, deviations in the dorsal part of the orbit with an elevation of the orbits were particularly responsible for this.

In contrast, the need for secondary revisions was even reduced from 11.1% in APOI to zero in PSI. In comparison with the study by Nikunen et al., in which the operative revision rates were recorded in a retrospective study of 232 operated patients and these were only between 3.8 and 6.5%, 11.1% appear to be somewhat higher and 0% significantly lower. It is interesting that no intraoperative imaging was performed in their collective. Thus, a secondary evaluation of the postoperative imaging of the 232 patients showed that implant misplacement was very frequent. 27.2% of the implant positions were subsequently considered to be less than ideal [27]. Especially in the group after secondary revision, the reconstruction result was considered poor in 60% and acceptable in only 27.2%.

In the authors’ view, secondary revisions in our collective were also reduced by the consistent use of intraoperative 3D imaging and the significant improvement in the image quality of the X-ray images produced by the second-generation 3D C-arms after 2015.

The evaluation of the deviations in the implant position showed clear differences in the APOI and PSI groups. In the APOI group, 20% deviated from the optimal implant position in both the medial and central areas of the implant. The most relevant deviations in the medial position were missing medial wall reconstructions with an undercorrection of around 6 mm in three patients. This leads to an increase in the orbital volume with possible consecutive enophthalmos. In the PSI group, however, the deviation in the medial implant position was significantly lower at 6.6% and 0% in the central implant position. The maximum undercorrection of the medial wall was only 2.2 mm. This, together with the fact that significantly more patients in the PSI group had a fracture of the medial wall, leads to the conclusion that reconstruction, especially of the medial wall, using a PSI is significantly more precise and easier.

The deviation in the dorsal part of the implant was even more obvious in the APOI group. 60% of the patients with standard implants showed a dorsal deviation. In 13.3%, this deviation was greater than 3 mm with a maximum of 6 mm overcorrection. This seems particularly relevant, as overcorrections around the dorsal orbital bone often seem to be accompanied by more clinical symptoms. In contrast, the PSI group had a deviation in the dorsal part of only 6.6% (*n* = 2). In particular, the deviation of 1.1 mm and 2.2 mm was much less than in the APOI group. This leads to the conclusion that the more precise positioning of the PSI in the anterior and lateral parts of the orbit clearly simplifies the positioning in the deep orbit.

Therefore, the use of patient-specific CAD/CAM orbital implants appears to be an advantage over anatomical preformed orbital implants in principle, but especially in more complex fractures [24,25,28].

In comparison with the literature, however, there were also rather large deviations for patient-specific orbital implants, which partly corresponded to the results for the standard implant in our study. The examination of the implant position of the PSI in a smaller case series of nine patients by Kormi et al. showed a maximum deviation of 4 mm anterolaterally, 2.4 mm anteromedially and of 4.8 mm at the posterior reference point [29]. The patient-specific implants used in this study appeared rather slightly smaller in size and mainly covered the bone defect. This could be an indication that the solution per se is not based in the PSI but mainly in the partially extended geometry of the PSI with a reproducible and clinically well intraoperatively verifiable lateral contact zone. This unique fitting individual geometry for the orbit can be described as a one-fit-design (OFD). However, this study also has its limitations. Due to the retrospective study design, a systematic recording of pre- and postoperative clinical findings of the patients was not possible. This would have been of particular interest to patients with major deviations. Whether there was a relevant enophthalmos in the absence of a medial wall reconstruction or motility restrictions in the case of significant overcorrection in the dorsal part cannot be verified with certainty. Additionally, the decisions that led to a revision of the orbit in the course of a second operation are only partially and not completely documented. It is unclear whether a retrospective evaluation of the intraoperative imaging, a new postoperative CT scan or the patient’s clinical symptoms ultimately led to the indication for revision surgery. Furthermore, the operations were performed by different surgeons. The operations carried out were performed by board-certified specialists and by experienced residents under supervision. A general dependence on the results, especially in the APOI group, seems unlikely, as most of the operations were performed by experienced specialists. According to the authors, it seems to be the case that more young specialists and senior residents performed operations in the PSI group. The further development of intraoperative 3D C-arms may also have influenced the results. In particular, the image quality and sharpness of detail were significantly improved in the newer generation devices. These new devices were increasingly used in the PSI group, as psi were inserted more frequently in the last five years. Although this may explain the lack of need for secondary reconstruction, the significantly lower intraoperative revision rates seem to be due to the better applicability of the PSI alone, independently of the 3D C-arm quality.

Another possible limitation results from the digital reconstruction of the orbital defects by mirroring the opposite side. There are differences in the morphology of the two orbits. These are based on physiological facial asymmetry. The method of mirroring was considered the gold standard for the last decade. Gass et al. were able to demonstrate a median error of 0.7 mm for small and 0.73 mm for large orbital defects. In this study, it was possible to achieve an almost exact anatomical reconstruction of the defective orbita by the method of mirroring, especially taking into account the s-shaped configurations of the orbital floor. In the future, even higher precision could be achieved by using statistical shape models [3].

Furthermore, this study did not compare the financial aspects of anatomical preformed implants and patient-specific CAD/CAM implants. The higher costs for the CAD/CAM implants and the possible slightly longer illness-related absence of the patient due to the rather delayed time of surgery appear to be primarily disadvantageous. On the other hand, in addition to the significantly reduced postoperative revision rate, there could also be possibly faster freedom from symptoms with a more precise implant position of the PSI. Further prospective (multicenter) studies could provide information on this.

## 5. Conclusions

We conclude that patient-specific implants are well suited for primary and secondary reconstruction of the orbit. Their use in combination with intraoperative imaging could significantly reduce secondary revision rates compared to anatomical preformed orbital implants. The intraoperative revision rate was also significantly lower with PSI. The evaluation of the precision of the implant position showed a significantly more accurate reconstruction in the medial and central orbit with PSI. In particular, the misalignment in the dorsal orbit, which frequently occurs when using standard implants, could be massively reduced with PSI. The OFD of the PSI appears to be significantly responsible for the reduction in revision rates.

## Figures and Tables

**Figure 1 jpm-13-00846-f001:**
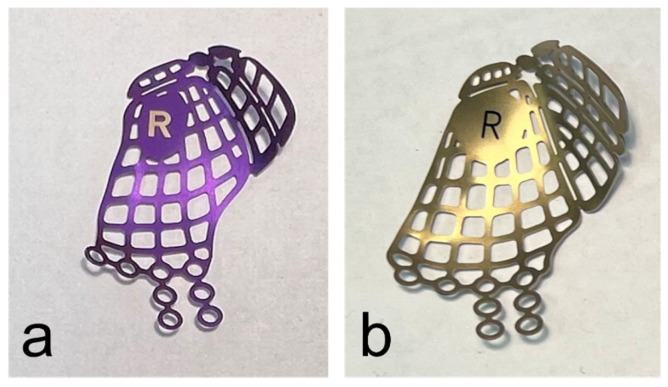
Anatomical preformed titanium meshes for the right orbita in small (**a**) and large (**b**) sizes.

**Figure 2 jpm-13-00846-f002:**
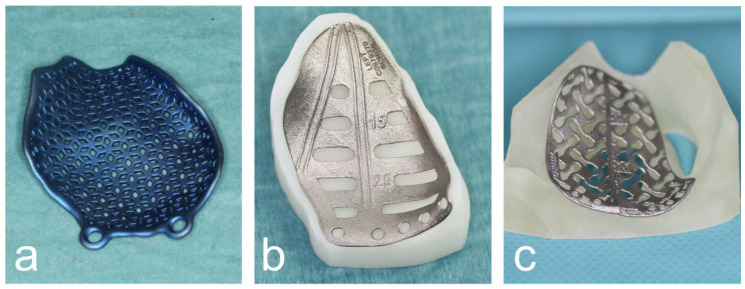
Various CAD/CAM patient-specific implants produced by DePuy Synthes© (**a**), KLS© (**b**) and ReOss© (**c**).

**Figure 3 jpm-13-00846-f003:**
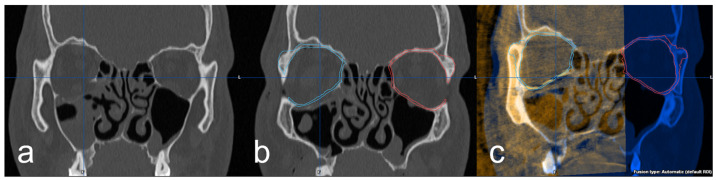
(**a**) coronal view of the right orbital floor fracture, (**b**) representation of the segmented intact left orbita (red) and mirroring to the right defect side (blue) and (**c**) fused intraoperative imaging with superimposition of the implant and the digitally reconstructed orbita (blue).

**Figure 4 jpm-13-00846-f004:**
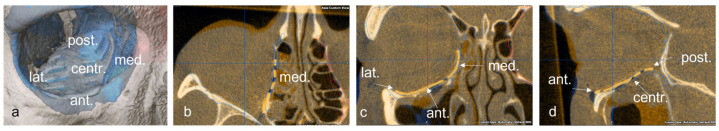
Representation of the five measuring points on the implant in the 3D reconstruction (**a**), and in the axial (**b**), coronal, (**c**) and sagittal (**d**) slices of the CT.

**Figure 5 jpm-13-00846-f005:**
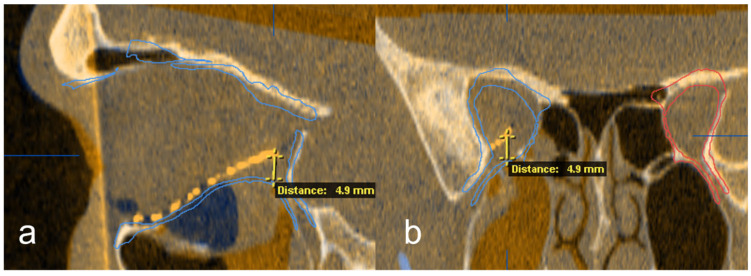
Exemplary deviation measurement at the posterior implant point to the digitally reconstructed orbit in the sagittal (**a**) and coronal (**b**) reconstruction out of the multiplanar view.

**Table 1 jpm-13-00846-t001:** States the deviations >1 mm in the APOI and de PSI-group with an indication of the mean values and the maximum value.

Deviation	APOI			PSI		
Medial	>1 mm 20%	Mean Value 3.17 mm	Maximum 6 mm	>1 mm 6.70%	Mean Value 1.65 mm	Maximum 2.2 mm
Anterior	4.4%	2.2 mm	2.3 mm	3.30%	1.1 mm	1.1 mm
Lateral	0%			0	0	
Posterior	60%	2.49 mm	6.7 mm	10%	1.76 mm	2.33 mm
Overall	66.7%	2.59 mm	6.7 mm	10%	1.54 mm	2.33 mm

**Table 2 jpm-13-00846-t002:** States intraoperative revisions and secondary operations in the APOI- and PSI-group.

	APOI	PSI
Intraoperative Revisions	26.60%	13%
Secondary Operations	11.10%	0%

## Data Availability

The datasets generated and analyzed during the current study are not publicly available due to institutional restrictions but are available from the corresponding author upon reasonable request.

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
