# Peer review of "Comparison of Anatomical Preformed Titanium Implants and Patient-Specific CAD/CAM Implants in the Primary Reconstruction of Isolated Orbital Fractures—A Retrospective Study"

_jpm, 2023, doi:10.3390/jpm13050846_

Round 1
Reviewer 1 Report
This observational retrospective single-center study on the comparison of anatomically preformed titanium implants and patient-specific CAD/CAM implants in the primary reconstruction of isolated orbital fractures gives the reader a wide overview of currently used diagnostic and therapeutic methods as well as a presentation of own experience in this field, based on the adequate number of cases.
In methodology, some clinical data on the degree of enophthalmos, range of diplopia, and restriction of eyeball motility in the pre-and postoperative period are missing. These data might have supported the decision on secondary surgical intervention. However, due to the retrospective design of the study, it is not possible to make the necessary corrections. The authors are fully aware of the limitations of their study and explain the reasons in the discussion section. Meticulous measurements and proper statistical tools enable objective results. The quality of the discussion is enhanced by well-chosen and up-to-date references.
I strongly recommend this paper for publication.
The overall quality of medical English is good. There are minor mistakes, e.g. "a anatomical" instead of "an anatomical" or "different CAD/CAM patient specific implants" instead of "Various... patient-specific...".
Author Response
Dear respected colleague, thank you very much for the very positive feedback. We are very pleased that you appreciate and value this study. In accordance with your recommendation, we will be happy to have this manuscript checked for linguistic errors with the help of a native speaker and correct these errors afterwards.
Thank you again for your time, your efforts and your improvements.
Kind regards
Sebastian Pietzka

Reviewer 2 Report
Re: jpm-2392194
Comparison of anatomical preformed titanium implants and patient-specific CAD/CAM implants in the primary reconstruction of isolated orbital fractures – a retrospective study
This study evaluated the accuracy of reconstruction of orbital fractures by using CAD/CAM implants and preformed implants. This is an important study because it proved the usefulness of CAD/CAM implants for reconstruction of the orbital fracture.
Some minor points that should be modified or added are as follows:
1. The left and right orbital regions do not have exactly the same morphology, although the healthy orbital region was mirrored to the affected side and the reconstruction implant was created based on that data. Therefore, it is necessary to provide details of the most important anatomical structures when superimposing the mirrored images.
2. The frequency of intraoperative repositioning and reoperations were shown. What were the guidelines for each?
3. In the Result section needs tables to make easier to understand for readers.
Author Response
Dear respected colleague, thank you very much for the feedback and the constructive improvements. We have gladly revised them in the manuscript. We appreciate your time and efforts, which have contributed to a significant improvement. We hope that the changes have been made to your satisfaction.
Ad 1.) According to your comment, this limitation has been further discussed in the text and supported with literature.
Another possible limitation results from digital reconstruction of the orbital defects by mirroring the opposite side. There are differences in the morphology of the two orbits. These are based on physiological facial asymmetry. The method of mirroring was considered the gold standard for the last decade. Gass et. al were able to demonstrate a median error of 0.7mm for small and 0.73mm for large orbital defects. In this study it was possible to achieve an almost exact anatomical reconstruction of the defective orbita by the method of mirroring, especially taking into account the s-shaped configurations of the orbital floor. In the future, even higher precision could be achieved by using statistical shape models [3].
Ad 2.) In accordance with your sensible comments, these two sections have been added to the manuscript for better comprehensibility.
“Secondary revision surgery was performed in all 5 patients within a mean of 4 days (median 5, min 1, max 6 days) due to significant postoperative restrictions of motility and double vision in the central field of the vision. In all these cases there was a clear elevation of the orbital grid in the posterior part.”
“Deviations between implant position and bone apposition have been described as an indication for intraoperative correction in the case of a supposedly clinically good position. According to the evaluation of the surgical reports, deviations in the dorsal part of the orbit with an elevation of the orbital meshes were particularly responsible for this. In contrast, the need for secondary revisions was even reduced from 11.1% in APOI to zero in PSI. “
Ad 3.) The following tables have been inserted in the results section to show both the differences and the corrections in the two groups more clearly.
Thank you again for your time, efforts and improvements. We hope that these improvements are in line with your expectations.
Kind regards
Sebastian Pietzka

Round 2
Reviewer 2 Report
Now my points were completely resolved. This article now achieved the level of the journal.